

# High resolution wave and hydrodynamics modelling in coastal areas: operational applications for coastal planning, decision support and assessment

Achilleas G. Samaras[1], Maria Gabriella Gaeta[1], Adrià Moreno Miquel[2], Renata Archetti[2]

[1]CIRI – EC, Fluid Dynamics Unit, University of Bologna, Via del Lazzaretto 15/5, Bologna, 40131, Italy
[2]Department of Civil, Chemical, Environmental and Materials Engineering, University of Bologna, Viale Risorgimento 2, Bologna, 40136, Italy

*Correspondence to*: A. G. Samaras (achilleas.samaras@unibo.it)

**Abstract.** Numerical modelling has become an essential component of today's coastal planning, decision support and risk assessment. High resolution modelling offers an extensive range of capabilities regarding simulated conditions, works and practices, and provides with a wide array of data regarding nearshore wave- and hydro- dynamics. In the present work, the open-source TELEMAC suite and the commercial software MIKE21 are applied to selected coastal areas of South Italy. Applications follow a scenario-based approach in order to study representative wave conditions in the coastal field; the models' results are intercompared in order to test both their performance and capabilities, and are further evaluated on the basis of their operational use for coastal planning and design. A multiparametric approach for the rapid assessment of wave conditions in coastal areas is also presented, and implemented in areas of the same region. The overall approach is deemed to provide useful insights on the tested models and the use of numerical models – in general – in the above context, especially considering that the design of harbours, coastal protection works and management practices in the coastal zone is based on scenario-based approaches as well.

## 1 Introduction

Accurate predictions of waves, currents and sea level variations in coastal areas are essential for a wide range of research and operational applications, as they govern inundation, sediment and pollutant transport, coastal morphology evolution and interactions with structures. Accordingly, numerical models that can serve the above purposes have become the main tool for researchers, engineers and policymakers around the world involved in coastal planning, risk management and monitoring activities.

Following the above considerations, the development of reliable modelling systems or methods that can scale down from the ocean to the coastal scale has emerged as a need in today's research. Reliable information on the hydrodynamics of the zone defined as nearshore, in particular, can serve a key role in coastal planning and hazard mitigation, as relevant processes at that scale differ significantly from the ones described in larger-scale oceanographic models. It is self-evident that, in the



above context, the capabilities and limitations of such systems and methods – apart from their structure – would depend on those of the numerical models they comprise.

A series of model coupling and nesting techniques, as well as entire methodological frameworks, have been proposed and applied in various research attempts for the development of modelling systems with the aforementioned characteristics.

Among the early works on the subject, one can indicatively refer to Ozer et al. (2000), who proposed a coupling module for tides, surges and waves based on the WAM-cycle 4 wave model (Komen et al., 1994) and a revised version of the hydrodynamics model of van den Eynde et al. (1995), applying it, however, to relatively low-resolution simulations for the North Sea. Regarding more recent and complete attempts, one can refer to the work of: Warner et al. (2010), who developed the Coupled Ocean–Atmosphere–Wave–Sediment Transport system ("COAWST"), using SWAN (Booij et al., 1999) and

the Community Sediment Transport Model (Warner et al., 2008) for the simulation of nearshore wave- and morpho-dynamics, respectively; Ge et al. (2013), who developed the FVCOM system (based on unstructured grid models) to simulate multi-scale dynamics at the East China Sea shelf and the Changjiang Estuary; and Barnard et al. (2014), who developed a modelling system for predicting storm impact on high-energy coasts ("CoSMoS"), using for the simulation of nearshore processes Delft3D-FLOW (Lesser et al., 2004; Delft Hydraulics, 2007), SWAN (Booij et al., 1999) and XBeach

(Roelvink et al., 2009).

On the other hand, integrated systems comprising atmosphere, ocean and coastal models do present a number of challenges for their users, regarding both data interoperability and downscaling/nesting techniques, also demanding significant computational expense in order to arrive to high resolution simulations near coasts. Furthermore, for a series of activities in coastal/marine planning (e.g. identification of wave energy sites, see: Reikard, 2009; Bozzi et al., 2014), vulnerability/risk

assessment (e.g. Stockdon et al., 2012; Idier et al., 2013) and coastal protection measures/infrastructure design (e.g. van Duin et al., 2004; Burcharth et al., 2014; Karambas, 2014; Karambas and Samaras, 2014), either only parts of local hydrodynamics information are required (mainly wave properties to drive nearshore models), or the respective approaches are based on the study of frequent/extreme condition scenarios. Accordingly, a number of methods have been developed in order to estimate coastal wave properties from offshore information or larger-scale simulations. One can refer to the early

work of O'Reilly and Guza (1993), who proposed wave energy transformation coefficients based on the comparison of two spectral wave models' results, or more recent ones using nesting and data assimilation schemes (Bertotti and Cavaleri, 2012; Rusu and Soares, 2014), and machine-learning techniques (Camus et al., 2011; Plant and Holland, 2011b, a). A work that stands out in recent literature is the one of Long et al. (2014), who proposed a probabilistic method based on model scenarios for constructing wave time-series at inshore locations.

The present work follows the rationale described right above, comparing two modelling suites in the representation of nearshore dynamics, and proposing a multiparametric scenario-based approach for the rapid assessment of wave conditions in coastal zones. Nevertheless, this work also served as the background study for the development of a modelling system coupling atmosphere, ocean and coastal dynamics, as described in Gaeta et al. (2016).



In the following, the open-source TELEMAC suite is compared with the well-known commercial software MIKE21 (developed by ©DHI Group) in fundamental wave-hydrodynamics modelling applications, aiming to test models' performance and representation of the various processes governing wave propagation and wave-induced nearshore hydrodynamics. The latter (i.e. MIKE21) is also used for the implementation of the aforementioned multiparametric

approach based on a trilinear interpolation algorithm. The study areas for the presented applications are all located in South Italy and comprise: the coastal area around the city/port of Brindisi, the coastal area around the city/port of Bari, and the Gulf of Taranto (the latter only for the multiparametric approach's applications). TELEMAC and MIKE21 results are compared on the basis of wave/current characteristics, along linear trajectories from the offshore to the nearshore and at specific points inside/outside the breaker zone and near harbour entrances (for the study area of Bari). As for the scenario-

based approach, its background and formulation are presented in detail, along with its implementation in the framework of an operational system, supporting the rationale behind this study and setting the basis for future work on the same path.

## 2 Methods

### 2.1 Wave and hydrodynamics modelling

Wave modelling within the TELEMAC and MIKE21 suites is performed using TOMAWAC and MIKE21-SW, respectively.

TOMAWAC and MIKE21-SW are characterized as *third-generation* spectral wave models, as they do not require any parameterization on either the spectral or the directional distribution of power (or action density). The physical processes modelled comprise: (a) energy source/dissipation processes (wind driven interactions with atmosphere, dissipation through wave breaking / whitecapping / wave-blocking due to strong opposing currents, bottom friction-induced dissipation), (b) non-linear energy transfer conservative processes (resonant quadruplet interactions, triad interactions), and (c) wave

propagation-related processes (wave propagation due to the wave group / current velocity, depth-/current- induced refraction, shoaling, interactions with unsteady currents). The models compute the evolution of wave action density $N$ by solving the action balance equation (Booij et al., 1999):

$$\frac{\partial N}{\partial t} + \nabla_{x,y}\left[(\vec{c}_g + \vec{U})N\right] + \frac{\partial}{\partial \sigma}(c_\sigma N) + \frac{\partial}{\partial \theta}(c_\theta N) = \frac{S_{tot}}{\sigma} \tag{1}$$

where $N = E/\sigma$ , $E$ being the variance density and $\sigma$ the relative angular frequency, $\vec{c}_g$ is the intrinsic group velocity vector,

$\vec{U}$ is the ambient current, $c_\sigma$ , $c_\theta$ are the propagation velocities in spectral space $(\sigma,\theta)$, and $S_{tot}$ is the source/sink term that represents all physical processes which generate, dissipate or redistribute energy. Broken down to its components, $S_{tot}$ can be written as:

$$S_{tot} = S_{in} + S_{wc} + S_{nl4} + S_{bf} + S_{br} + S_{nl3} \tag{2}$$





where $S_{in}$ represents the energy transfer from wind to waves, $S_{wc}$ the dissipation of energy due to whitecapping, $S_{nl4}$ the nonlinear transfer of energy due to quadruplet (four-wave) interactions, $S_{bf}$ the dissipation due to bottom friction, $S_{br}$ the dissipation due to wave braking, and $S_{nl3}$ the nonlinear transfer of energy due to triad (three-wave) interactions. TOMAWAC and MIKE-SW parameterize similarly the above processes; TOMAWAC, however, does offer more options

regarding the available approaches/models to be used for most of them. Therefore, and regarding the processes of interest for the model intercomparison as presented in Section 3.1.3, the respective common approaches/models applied in this work are: the Battjes and Janssen (1978) model for bathymetric breaking; the model of Hasselmann et al. (1973) for bottom friction dissipation using a constant friction coefficient; the Komen et al. (1984) and Janssen (1991) dissipation model for whitecapping; and the LTA (Lumped Triad Approximation) model of Eldeberky and Battjes (1983) for triad interactions (the

SPB model of Becq, 1998 – available only in TOMAWAC – is also tested). As for diffraction, its effect is simulated using the phase-decoupled approach proposed by Holthuijsen et al. (2003), based on the revised version of the Mild Slope Equation model of Berkhoff (1972), proposed by Porter (2003). Both models solve the governing equation by means of finite element-type methods to discretize geographical and spectral space, while the geographical domain is discretized by unstructured triangular meshes.

Hydrodynamics modelling within the TELEMAC and MIKE21 suites is performed using TELEMAC-2D and MIKE21-HD, respectively. The models solve the 2D shallow water equations (also referred to as Saint-Venant equations; see Hervouet, 2007), derived by integrating the Reynolds-averaged Navier-Stokes equations over the flow depth. Adopting the formulation of TELEMAC-2D for Cartesian coordinates, the equations of continuity and momentum along the x- and y- axes can be written as Eqs. (3), (4) and (5), respectively:

$$\frac{\partial h}{\partial t} + \vec{u} \cdot \vec{\nabla}(h) + h\,div(\vec{u}) = S_h \qquad (3)$$

$$\frac{\partial u}{\partial t} + \vec{u} \cdot \vec{\nabla}(u) = -g\frac{\partial \zeta}{\partial x} + S_x + \frac{1}{h}div\left(h v_t \vec{\nabla}u\right) \qquad (4)$$

$$\frac{\partial v}{\partial t} + \vec{u} \cdot \vec{\nabla}(v) = -g\frac{\partial \zeta}{\partial y} + S_y + \frac{1}{h}div\left(h v_t \vec{\nabla}v\right) \qquad (5)$$

where $h$ is the water depth, $u,v$ are the velocity components and $\vec{u}$ the velocity vector, $g$ is the gravitational acceleration, $v_t$ is the momentum diffusion coefficient, $\zeta$ is the free surface elevation, $S_h$ is a term representing sources/sinks of fluid, and $S_x$, $S_y$

are terms representing sources/sinks of momentum within the domain (i.e. wind, Coriolis force, bottom friction). These primitive equations are solved by means of finite element/volume methods, while the geographical domain is discretized by unstructured triangular meshes. As also mentioned in the previous for TOMAWAC and MIKE-SW, and although TELEMAC-2D and MIKE21-HD have a lot of similarities, TELEMAC-2D does offer more parameterization options regarding the definition of physical and numerical parameters. In the present work, the use of the hydrodynamics models is



focused on the representation of wave-generated currents, a task achieved through their direct coupling with the respective spectral wave models within the TELEMAC and MIKE21 suites.

## 2.2 Multiparametric approach for the rapid assessment of nearshore wave conditions

The methodology followed in the present work for the rapid assessment of nearshore wave conditions (within the framework set in the previous; see Section 1), comprises a number of steps aiming to establish an efficient and computationally-reasonable approach for operational use. The approach is scenario-based, thus its first step consists in defining a number of scenarios representing wave conditions in the wider area of interest. This is done by performing a spectral analysis of sea surface elevation records from nearshore/offshore buoys in order to produce a dataset of three aggregated wave parameters, namely: the significant wave height $H_s$, the peak period $T_p$ and the mean wave direction $Dir_m$. Next, dataset parameters are further divided into a number of classes each, forming by aggregation the sets of $H_s$ - $T_p$ - $Dir_m$ henceforth referred to as "scenarios". These scenarios are afterwards used (in sequence) as boundary conditions for the wave model runs, resulting in an extensive dataset of model results for the entire computational domain, stored in ASCII files, properly named on the basis of the input wave scenarios and formatted in columns according to the following convention (each row referring to a different node of the computational mesh): [Col.1= Node Latitude; Col.2= Node Longitude; Col.3= Water depth; Col.4= $H_s$; Col.5= $T_p$; Col.6= $Dir_m$]. These files form the high-resolution wave condition database along with a query algorithm, serving as the "bridge" between coarser-resolution operational models and the aforementioned produced dataset. The query algorithm is responsible for: (a) identifying the boundary wave conditions given by the coarser resolution model (as sets of $H_s$ - $T_p$ - $Dir_m$); and (b) scanning the dataset for the ASCII file corresponding to the specific wave conditions and retrieving it. In the – admittedly most probable – case that no dataset file matches exactly the set of defined wave parameters, the algorithm will additionally: (c) define the upper and lower classes' boundaries for all three parameters (i.e. $H_s$, $T_p$, $Dir_m$) on the basis of their original query values, scan the dataset and retrieve the respective ASCII files; (d) implement a trilinear interpolation in the three-dimensional $H_s$ - $T_p$ - $Dir_m$ space (according to Bourke, 1999; Kreyszig, 2010) for each node of the computational mesh; and finally (e) store the derived parameter values in a new query-tailored ASCII file. The latter will represent the nearshore wave conditions for the query-defined set of wave parameters.

It should be noted that the division to a large number of parameter classes at the first steps of this approach will lead to a large number of scenarios and, consequently, a large number of runs to be performed by the coastal wave model, with the respective effect on computational cost. However, this will accordingly lead to a higher accuracy of the trilinear interpolation method as well, considering that its intrinsic error becomes lower with the increase in scenario discretization. Given that – in the framework of an operational system – response speed is of the essence, the combination of the specific interpolation method with a adequately high number of defined scenarios is deemed to deliver the best performance overall due to its simplicity and implementation speed.





## 3 Application setup

### 3.1 Model intercomparison

#### 3.1.1 Conceptual approach

TELEMAC and MIKE21 have been extensively used over the years in research, operational and engineering design applications in maritime/coastal hydraulics; for MIKE21 this use leans significantly towards the last two categories, it being one of the most widespread used commercial suites for relevant applications. Their models have been separately evaluated and validated for several case studies. Regarding TELEMAC, exemplary reference can be made to the work of: Brière et al. (2007), on assessing its performance for a hydrodynamic case study; Brown and Davies (2009), Luo et al. (2013) and Villaret et al. (2013), on coupled wave – hydrodynamics – sediment transport / morphological modelling; Sauvaget et al. (2000) on the modelling of tidal currents; and Jia et al. (2015) on wave-current interactions in a river and wave dominant estuary. Regarding MIKE21, respective literature review would include the work of: Siegle et al. (2007) and Ranasinghe et al. (2010) on coupled wave – hydrodynamics – sediment transport / morphological modelling (the first focused on inlet morphodynamics and the second on morphological response to technical works); Babu et al. (2005) on the modelling of tide-driven currents; Kong (2014) on the impact of tidal waves on storm surge; and Aboobacker et al. (2009) and Arı Güner et al. (2013) on wave modelling. However, and given the fact that regarding system architecture and modelling components TELEMAC and MIKE21 have a lot of similarities (see also Section 2.1), literature has to show limited references on their comparative evaluation.

The rationale behind the model intercomparison presented in the following derives from the general framework within which this work is carried out, that is the use of high resolution wave and hydrodynamics models for: (a) the development and application of a multiparametric approach for the rapid assessment of wave conditions at inshore locations (presented in Sections 2.2 and 3.2); and (b) the development of a modelling system coupling atmosphere, ocean and coastal dynamics (presented in Gaeta et al., 2016). Accordingly, the TELEMAC and MIKE21 suites are compared in fundamental wave-hydrodynamics modelling applications, aiming to test models' performance and representation of the various processes governing wave propagation and wave-induced nearshore hydrodynamics. The comparison is performed for both single wave events and time-series or random waves, representative of typical applications for coastal planning, decision support and assessment. Apart from a coastal stretch near the city and harbour of Brindisi, applications (using only TOMAWAC) are also performed for the area around the city of Bari, including its harbour. Specifically regarding the latter – and given the inherent limitations the inclusion of the representation of diffraction poses on the results of phase-averaged models – it should be noted that the objective was solely to test the extent to which spectral models like TOMAWAC could be used to capture diffraction effects near harbour entrances (when the detailed agitation inside the harbour is not of interest), without the need to resort to separate time-demanding applications using phase-resolving models. The intercomparison also retains a strong user-oriented component, presenting examples of how models perform under typical coastal application scenarios and how basic physical processes affect the computed parameters of interest.



### 3.1.2 Study areas and mesh generation

The first of the two study areas is located northwest of the city of Brindisi (South Italy), comprising Torre Guaceto, a Marine Protected Area (MPA) and State Natural Reserve of significant importance. The selected rectangular outline of the domain for the model applications measures about 21km in the longshore and 7.5km in the cross-shore direction; Fig. 1(a) shows the

wider study area and the aforementioned outline. The second study area comprises the coastal area around the city and harbour of Bari (South Italy); the outline of the computational domain in this case measures about 16.5km in the longshore and 8.5km in the cross-shore direction (see Fig. 1(b)).

As mentioned in Section 2.1, both the TELEMAC and MIKE21 modelling suites discretize the computational domain by unstructured triangular meshes. Mesh generation for TELEMAC applications was done using Blue Kenue, a data

preparation, analysis and visualization tool for hydraulic modellers developed by the National Research Council of Canada; the respective work for MIKE21 was done using MIKE Zero, the DHI tool for managing MIKE projects.

The bathymetric and shoreline data used in this work resulted from the digitization of nautical charts acquired from the Italian National Hydrographic Military Service ("Istituto Idrografico della Marina Militare"). For the case study of Brindisi – Torre Guaceto the triangular mesh was created defining two density zones (20m edge length below the -10m isoline and

250m for the rest of the field), resulting in a mesh consisting of 55,340 nodes forming 109,124 elements. It should be noted that the mesh was first created in Blue Kenue and afterwards properly transformed to MIKE Zero format, maintaining the exact same nodes and connections in order to exclude mesh-dependent divergences in the model runs. Figure 1(c) shows the mesh and bathymetry of the computational domain, along with the three linear trajectories and six points for which model results will be intercompared (see Section 3.1.3). For the case study of Bari, three density zones were defined arriving to the

finest discretization of 10m edge length in order to represent harbour structures, 250m being the lowest discretization moving offshore. The resulting mesh consists of 25,202 nodes forming 46,144 elements; Fig. 1(d) shows the mesh and bathymetry of the computational domain, along with the linear trajectory and three points used for results' analysis (see Section 3.1.3).

### 3.1.3 Application setup for model intercomparison

Table 1 presents a detailed overview of all model runs; the table is divided in two parts, the top one referring to the Brindisi – Torre Guaceto applications and the bottom one to the Bari applications (see also Fig. 1). Runs for the Brindisi – Torre Guaceto case study refer to coupled wave and hydrodynamics models applications, that is: coupled TOMAWAC – TELEMAC-2D and MIKE21-SW – MIKE21-HD runs for the TELEMAC and MIKE21 suites, respectively. Runs for the Bari case study refer to standalone TOMAWAC applications, in the framework of the conceptual approach as presented in

Section 3.1.1. Every model run is assigned a different codename, henceforth used for its identification, with each line of Table 1 defining: the forcing used (i.e. single wave events or time-series of random waves); the processes included in the wave models' setup (see Table 2 and Section 2.1); the trajectories along which or the points at which results are





intercompared; the parameters included in the comparison; and, finally, a reference to the figure(s) presenting the specific results in Section 4.

The forcings were selected to represent a wide range of conditions regarding the wave climate in the areas of interest. The two single wave events selected, henceforth denoted as *WE1* and *WE2*, represent the 50-year and 2-year return period waves

as resulted from the analysis of Regione Puglia (2009). The two 12-hour time-series selected, henceforth denoted as *TS1* and *TS2*, were identified after analysis of wave data from the buoy of Monopoli (lat/lon: 40°58.5' N / 17°22.6' E, depth: 90m), part of the Italian wave metric network "RON" ("Rete Ondametrica Nazionale"; Corsini et al., 2006). All their characteristics are presented in Fig. 2. The processes included in the wave models' setup are presented in Section 2.1.

Considering that model results presented over the entire computational domain (as 2D fields of the respective parameters)

would pose significant challenges to the perceptibility of any intercomparison attempt (between both different modelling suites and different processes), it was deemed preferable to compare model results along linear trajectories from the offshore computational boundary to the shoreline (for *WE1* and *WE2*) or at specific points (for *TS1* and *TS2*).  For the Brindisi – Torre Guaceto case study, trajectories *TRc1*, *TRc2* and *TRc3* were defined in order to capture areas of different/ representative bathymetry profiles alongshore; the pairs of points *PTc1 - PTc4*, *PTc2 - PTc5* and *PTc3 - PTc6* were defined at specific

locations of the aforementioned trajectories, respectively. The first point of each of the previous pairs was selected to fall within the breaker zone and second one before the breaker line; given that – regarding the hydrodynamics – the objective was to compare wave-generated currents, the hydrodynamics models' results were analyzed only at points *PTc1*, *PTc2* and *PTc3*. Points' locations were decided to not change between runs for different forcings, in order to facilitate the comprehensibility of the presented results. For the Bari case study, the objective being to test the diffraction algorithm's

performance in spectral wave models (see also Section 3.1.1), one trajectory was defined (*TRh1*) and three points along it: one at the vicinity of the outer breakwater tip (*PTh1*), one right at the middle of the harbour's entrance (*PTh2*), and one inside the harbour close to the entrance (*PTh3*). All trajectories, points and bathymetric profiles are presented in Figs. 1(c) and 1(d).

## 3.2 Multiparametric approach for the rapid assessment of wave conditions

The multiparametric approach presented in this work was applied to three areas of interest in South Italy:  the areas around the cities/ports of Brindisi and Bari, as well as the entire Gulf of Taranto (see Fig. 3). Accordingly, the scenarios representing wave conditions in the wider area were defined based on the analysis of data from the buoys of Monopoli (lat/lon: 40°58.5' N / 17°22.6' E, depth: 90m; see Fig. 3) and Crotone (lat/lon: 39°01.4' N / 17°13.2' E, depth: 95m; see Fig. 3), covering the period from January 1st, 1989 to December 31st, 2012. For each buoy dataset, wave parameters were further divided into a

number of classes each – according to Table 3 – forming by aggregation the scenarios (i.e. sets of $H_s$ - $T_p$ - $Dir_m$) to be used for the wave model runs. Figs. 4(a) and 4(b) show the frequencies of occurrence of the scenarios' $H_s$ - $T_p$ and $H_s$ - $Dir_m$ pairs, respectively, for the Monopoli dataset; Figs. 4(c) and 4(d) show the respective frequencies for Crotone. It should be noted





that all directions follow the nautical direction convention; negative values were used in Fig. 4(b) for representation issues, as gaps in certain direction ranges (i.e. corresponding to what would be seaward wave origins) were omitted.

Simulations were performed using MIKE-SW, the spectral wave model of the MIKE21 suite (see description in Section 2.1). Mesh generation was done using MIK Zero (Fig. 3 shows the modelling domains' outlines); the overall setup methodology

followed the one presented in Section 3.1, including the processes of energy dissipation due to bathymetric breaking and bottom friction. The previously defined scenarios were used – in sequence – as boundary conditions for the model runs; the scenarios resulted from the Monopoli dataset were used in the Brindisi and Bari runs, while the ones from the Crotone dataset in the Gulf of Taranto runs. Model results created three extensive datasets (one for each study area), stored in properly named and formatted ASCII files, as described in Section 2.2. The performance of the developed query algorithm,

also described in Section 2.2, was tested for a series of exemplary cases before its operational implementation.

In the framework of the Research Project "TESSA" (Development of Technologies for the Situational Sea Awareness), the specific multiparametric approach was applied using WAVEWATCH III (Tolman, 2009) as the coarser-resolution model that would feed sets of $H_s$ - $T_p$ - $Dir_m$ to the query algorithm in order to retrieve/create the nearshore wave conditions file based on MIKE-SW results; the model's rectilinear grid is presented in Fig. 3.

## 4. Results and discussion

As described in Section 3.1 and presented in Tables 1 and 2, model intercomparison regards the Brindisi – Torre Guaceto case study. Figs. 5 and 6 show the comparison of TELEMAC and MIKE21 results ($H_s$ - $T_m$ - $Dir_m$) along trajectories $TRc1$, $TRc2$ and $TRc3$ for forcing $WE1$, as well as the effect of different processes on $H_s$ along the specific trajectories, separately for each modelling suite; Figs. 7 and 8 show the respective results for forcing $WE2$. The overall agreement between model

results is good, and all parameters are very close for the majority of runs for both forcings, with a general observation being that TELEMAC constantly produces slightly higher values of $H_s$ and lower values of $T_m$ than MIKE21. The extensive set of runs tested (i.e. combinations of processes included in the wave models' setup) allows for a more detailed analysis of models' performance, as presented in the following. For runs $Tc11$ and $Tc21$, including the processes of breaking and bottom friction dissipation, $H_s$ values are practically overlapping along most part of all three trajectories, with the exception of the

divergences observed at the vicinity of the breaker line (more noticeable for the relatively mild slope $TRc1$ rather than $TRc2$ and $TRc3$); $T_m$ and $Dir_m$ show small divergences as well, mostly noticeable after breaking for the steeper slope profiles of $TRc2$ and $TRc3$ and for the higher-wave forcing $WE1$ (i.e. $Tc11$ run). The inclusion of the process of energy dissipation due to whitecapping in runs $Tc12$ and $Tc22$ results in a small decrease of $H_s$ overall, which is more clearly noticeable in Figs. 6(b) and 8(b) presenting such effects separately for TELEMAC and MIKE21; changes in $T_m$ and $Dir_m$ are barely noticeable

between $Tc11 - Tc12$ and $Tc21 - Tc22$ runs. The additional inclusion of the non-linear triad interactions in runs $Tc13$ and $Tc23$ leads to the most noticeable discrepancies between model results (again, more noticeable for the relatively mild slope $TRc1$ rather than $TRc2$ and $TRc3$), limited of course to the shallow water sections of the studied profiles/trajectories where

the specific process's effect becomes significant. Although both suites use the LTA model of Eldeberky and Battjes (1983), the inclusion of triads seems to have a rather small effect on MIKE21 $H_s$ results (slight decrease of wave height and shift of the breaker line seaward), with the effect on the wave energy spectrum, however, becoming more evident when comparing $T_m$ values. On the other hand, TELEMAC runs result in higher $H_s$ values right before breaking and quite lower $T_m$ values

inshore. $Dir_m$ results show small divergences for both modelling suites. Additionally to runs $Tc13$ and $Tc23$, the effect of triad interactions was also tested using the the SPB model of Becq (1998), available as an alternative option only in TOMAWAC; the test was included as $Tc14$ in the set of runs, and its results are represented as dotted lines in all figures (noted accordingly). Following Becq-Girard et al. (1999) remarks on the validity range of the LTA model, $Tc14$ results show indeed a quite different representation of the process by TOMAWAC, with milder evolution of the wave energy onshore and

smaller changes to all parameter values than $Tc13$ produced (see Figs. 6(b) and 8(b) in particular).

Figures 9 and 10 show the comparison of TELEMAC and MIKE21 results ($H_s$ and *Curr. speed/direction*, respectively), for the time-series forcing *TS1*; Figs. 11 and 12 show the respective results for forcing *TS2*. Significant wave height values are compared at points along trajectories *TRc1*, *TRc2* and *TRc3* (see Fig. 1), three of them within the breaker zone (*PTc1*, *PTc2*, *PTc3*) and three outside of it (*PTc4*, *PTc5*, *PTc6*); the wave generated currents' speed and direction are compared only at

points *PTc1*, *PTc2* and *PTc3* (see also Section 3.1.3). Regarding $H_s$, the comparison between results at pairs *PTc1-PTc4*, *PTc2-PTc5* and *PTc3-PTc6* highlights the effect different processes have on model results for propagating waves towards the nearshore and how including/omitting them may become significant (or insignificant) for various operational, planning and engineering design applications in coastal areas. TELEMAC and MIKE21 results at points *PTc4*, *PTc5* and *PTc6* are close and in-phase for all processes, with higher discrepancies observed for the higher-waves forcing *TS2*. At points *PTc1*, *PTc2*

and *PTc3*, the conclusions drawn from the analysis of the wave events' results in the previous can be clearly identified here as well, with the most significant alterations in the different suites' results observed again for the runs where triad interactions were included in the modelled processes (i.e. *Tc33/Tc34* and *Tc43/Tc44*); it should be also noted that the higher-wave forcing *TS2* leads to smaller variations of $H_s$ than *TS1* overall, thus minimizing the effect of the different approach for triads modelling in run *Tc44* too. Regarding the wave-generated currents, TELEMAC and MIKE21 results are in relatively

good agreement for all runs considering the order of magnitude of the resulting current speeds, as well as the sensitivity of current directions within the breaking zone. Figure 10 shows that for runs *Tc31* and *Tc32* results are very close with the exception of the period up to hour 4 at *PTc1*, where TELEMAC shows current speeds close to zero (with the respective effect on current direction). As noted in the previous, the introduction of triad interactions results in a more significant effect when modelled with TELEMAC, although the SPB model does lead to smoother results regarding both speed and direction

(run *Tc34*). Point *PTc3* shows larger divergences than points *PTc1* and *PTc2*, that attributed to the combination of its location in the computational domain and the significant shift in the forcing's direction after hour 6 (see Fig. 2). Figure 12 shows that at points *PTc1* and *PTc2* results are in good agreement for both TELEMAC and MIKE21, following the observation regarding the small $H_s$ variations observed in the breaking zone for *TS2* (see Fig. 11). At *PTc3* TELEMAC



results are similar to the MIKE21 ones between hours 3 and 7, but significantly higher at the beginning and the end of the simulated time series.

Regarding the Bari case study, it should be stated again (as in Section 3.1.1) that the objective of its inclusion in this work was solely to test the extent to which spectral models could be used to capture diffraction effects near harbour entrances in

the framework of operational approaches like the one presented in Section 3.2. That, keeping in mind the inherent limitations the representation of diffraction poses on the results of phase-averaged models, as well as the fact that a detailed study of harbour agitation would require the use of a phase-resolving model. Figure 13 shows TOMAWAC results (with and without the inclusion of diffraction) along trajectory *TRh1* and as wave fields at the area of the harbour, for forcings *WE1* (Figs. 13(a), (b) and (c), respectively) and *WE2* (Figs. 13(d), (e) and (f), respectively). Results show noticeable differences in wave

characteristics around the breakwater's tip and near the harbour entrance, while the model also manages to capture the diffusion of the wave height inside the harbour area; larger effects are observed for the higher-wave forcing *WE1*. Figure 14 shows TOMAWAC results (with and without the inclusion of diffraction) at points *PTh1*, *PTh2*, *PTh3* for forcings *TS1* and *TS2* (Fig. 14(a) and 14(b), respectively). Differences are noticeable for all parameters, being relatively more significant at points *PTh2 - PTh3* and for the higher-waves forcing *TS2*.

Finally, the multiparametric approach presented in this work was successfully implemented in the framework of the Italian Flagship Research Project "TESSA", using WAVEWATCH III (Tolman, 2009) to feed sets of offshore wave characteristics to the query algorithm in order to provide with the nearshore wave conditions from the created database of MIKE-SW results. Its performance was tested for a series of different wave conditions for the three areas of interest (i.e. Brindisi, Bari, and Gulf of Taranto; see Fig. 3) and the algorithm managed to deliver results in a fast and seamless way at all times.

**5. Conclusions**

This work presents the comparison of the TELEMAC and MIKE21 modelling suites in fundamental wave and hydrodynamics applications for the representation of nearshore dynamics, and proposes a multiparametric scenario-based approach for the rapid assessment of wave conditions in coastal zones that aims to serve as an operational tool for coastal planning, decision support and assessment. The study areas for the presented applications are all located in South Italy and

comprise: the coastal area around the city/port of Brindisi, the coastal area around the city/port of Bari, and the Gulf of Taranto. For the first one, TELEMAC and MIKE21 are intercompared for a series of application setups aiming to test models' performance and representation of the various processes governing wave propagation and wave-induced nearshore hydrodynamics. For the study area of Bari (including its harbour), the spectral wave model of TELEMAC (i.e. TOMAWAC) is applied with and without the inclusion of the representation of the processes of diffraction, in order to test the extent to

which similar models could be used to capture diffraction effects near harbour entrances, in the framework of operational approaches like the one presented in this work when the detailed agitation inside the harbour is not of interest. TELEMAC and MIKE21 results are compared on the basis of wave/current characteristics, along linear trajectories from the offshore to





the nearshore and at specific points inside/outside the breaker zone and near the entrance of the harbour for the study area of Bari. Analysis shows an overall satisfactory agreement between the two modelling suites and is deemed to provide useful insights on both their individual capabilities and their comparative evaluation. The specific tasks also served as the background study for the development of a modelling system based on a multiple-nesting approach, coupling atmosphere,

ocean and coastal dynamics (described in Gaeta et al., 2016), while retaining a strong user-oriented component, showing examples of how models perform under typical coastal application scenarios and how basic physical processes affect the computed parameters of interest. The proposed multiparametric approach is presented in detail, consisting of: the definition of a number of wave scenarios on the basis of field measurements, a dataset of wave model results using these scenarios as boundary conditions, and a query algorithm based on the trilinear interpolation that bridges coarser-resolution operational

models and the aforementioned dataset in order to provide query-tailored fields of nearshore wave dynamics. The implementation of the specific approach as part of an operational chain for all three study areas in South Italy in the framework of the Italian Flagship Project "TESSA" supports the rationale behind this study, while setting the basis for future work on the same path.

**Acknowledgments**

This work was performed and funded in the framework of the Italian Flagship Project ''TESSA – Development of Technologies for the Situational Sea Awareness'' supported by the PON01_02823/2 "Ricerca & Competitività 2007–2013" program of the Italian Ministry for Education, University and Research.

The authors would like to thank Dr. Andrea Pedroncini from DHI Italia and Dr. Antonio Bonaduce from the Euro-Mediterranean Center on Climate Change for their valuable help on various modelling aspects of the MIKE21 suite and on

assisting with the operational implementation of the multiparametric approach in the framework of the TESSA Project, respectively.

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





Table 1. Overview of TELEMAC and MIKE21 model runs.

**Brindisi – Torre Guaceto**

| Run | Forcing | Processes | Comparison along/at | Compared parameters | Figure(s) |
|---|---|---|---|---|---|
| Tc11 |  | PRc1 |  |  |  |
| Tc12 | WE1 | PRc2 | TRc1, TRc2, TRc3 | $H_s$, $T_m$, $Dir_m$ | Figs. 5, 6 |
| Tc13 |  | PRc3 |  |  |  |
| Tc14[1] |  | PRc4 |  |  |  |
| Tc21 |  | PRc1 |  |  |  |
| Tc22 | WE2 | PRc2 | TRc1, TRc2, TRc3 | $H_s$, $T_m$, $Dir_m$ | Figs. 7, 8 |
| Tc23 |  | PRc3 |  |  |  |
| Tc24[1] |  | PRc4 |  |  |  |
| Tc31 |  | PRc1 |  |  |  |
| Tc32 | TS1 | PRc2 | PTc1, PTc2, PTc3, | $H_s$ | Fig. 9 |
| Tc33 |  | PRc3 | PTc4, PTc5, PTc6 | Curr. speed/direction[3] | Fig. 10 |
| Tc34[1] |  | PRc4 |  |  |  |
| Tc41 |  | PRc1 |  |  |  |
| Tc42 | TS2 | PRc2 | PTc1, PTc2, PTc3, | $H_s$ | Fig. 11 |
| Tc43 |  | PRc3 | PTc4, PTc5, PTc6 | Curr. speed/direction[3] | Fig. 12 |
| Tc44[1] |  | PRc4 |  |  |  |

**Bari[4]**

| Run | Forcing | Processes | Comparison along/at | Compared parameters | Figure |
|---|---|---|---|---|---|
| Th1 | WE1 | PRh1 | TRh1 | $H_s$, $T_m$, $Dir_m$ | Fig. 13 |
| Th1D |  | PRh2 |  |  |  |
| Th2 | WE2 | PRh1 | TRh1 | $H_s$, $T_m$, $Dir_m$ | Fig. 13 |
| Th2D |  | PRh2 |  |  |  |
| Th3 | TS1 | PRh1 | PTh1, PTh2, PTh3 | $H_s$, $T_m$, $Dir_m$ | Fig. 14 |
| Th3D |  | PRh2 |  |  |  |
| Th4 | TS2 | PRh1 | PTh1, PTh2, PTh3 | $H_s$, $T_m$, $Dir_m$ | Fig. 14 |
| Th4D |  | PRh2 |  |  |  |

[1] TELEMAC-only run (see Sections 2.1 and 3.1.1)

[2] $H_s$ = significant wave height, $T_m$ = mean wave period, $Dir_m$ = mean wave direction

[3] Current speed/direction are intercompared only at *PTc1*, *PTc2*, *PTc3*

5   [4] Standalone TOMAWAC runs (see Sections 2.1 and 3.1.1)





Table 2. Definition of the processes included in TELEMAC and MIKE21 spectral wave models' setup (see Table 1).

| Brindisi – Torre Guaceto | | | | | |
|---|---|---|---|---|---|
| Processes | Breaking | Bottom friction | Whitecapping | Triads (LTA) | Triads (SPB) |
| PRc1 | ● | ● | | | |
| PRc2 | ● | ● | ● | | |
| PRc3 | ● | ● | ● | ● | |
| PRc4[1] | ● | ● | ● | | ● |

| Bari[2] | | |
|---|---|---|
| Processes | Breaking | Bottom friction | Diffraction |
| PRh1 | ● | ● | |
| PRh2 | ● | ● | ● |

[1] Processes applied only to TELEMAC runs as *Triads (SPB)* is available only in TOMAWAC (see Sections 2.1 and 3.1.1)

[2] Processes applied to standalone TOMAWAC runs (see Sections 2.1 and 3.1.1)

5  Table 3. Class properties applied to the wave parameter datasets for scenarios' definition.

| Parameter | Minimum | Maximum | Class step |
|---|---|---|---|
| $H_s$ [m] | 0.1 | 6 | 0.1 |
| $T_p$ [sec] | 1.5 | 12 | 0.5 |
| $Dir_m$ [deg] | 0 | 355 | 5 |



Figure 1. Satellite images of the wider areas, outlines of the computational domains, meshes, bathymetries, linear trajectories and points for results' analysis for the Brindisi – Torre Guaceto ((a) and (c)) and Bari ((b) and (d)) case studies (background images from Google Earth, 2016; privately processed).



Figure 2. Characteristics of the wave events (*WE1*, *WE2*) and time-series (*TS1*, *TS2*) used as forcings for TELEMAC and MIKE21 runs (see also Table 1).

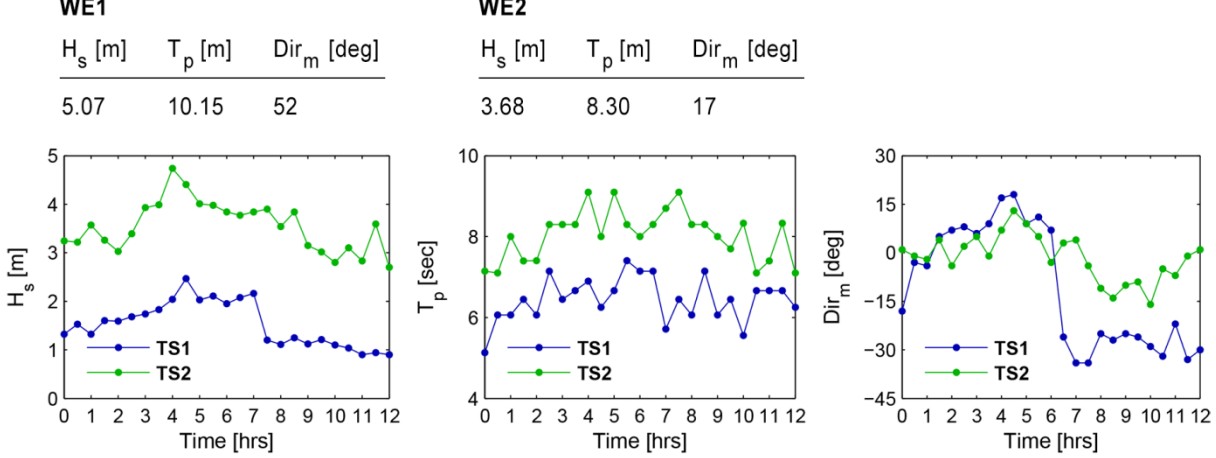





Figure 3. Computational domain outlines for the three areas in South Italy where the proposed multiparametric approach was applied, and locations of the Monopoli and Crotone buoys; the grid lines and points represent the WAVEWATCH III rectilinear grid (background image from Google Earth, 2016; privately processed).

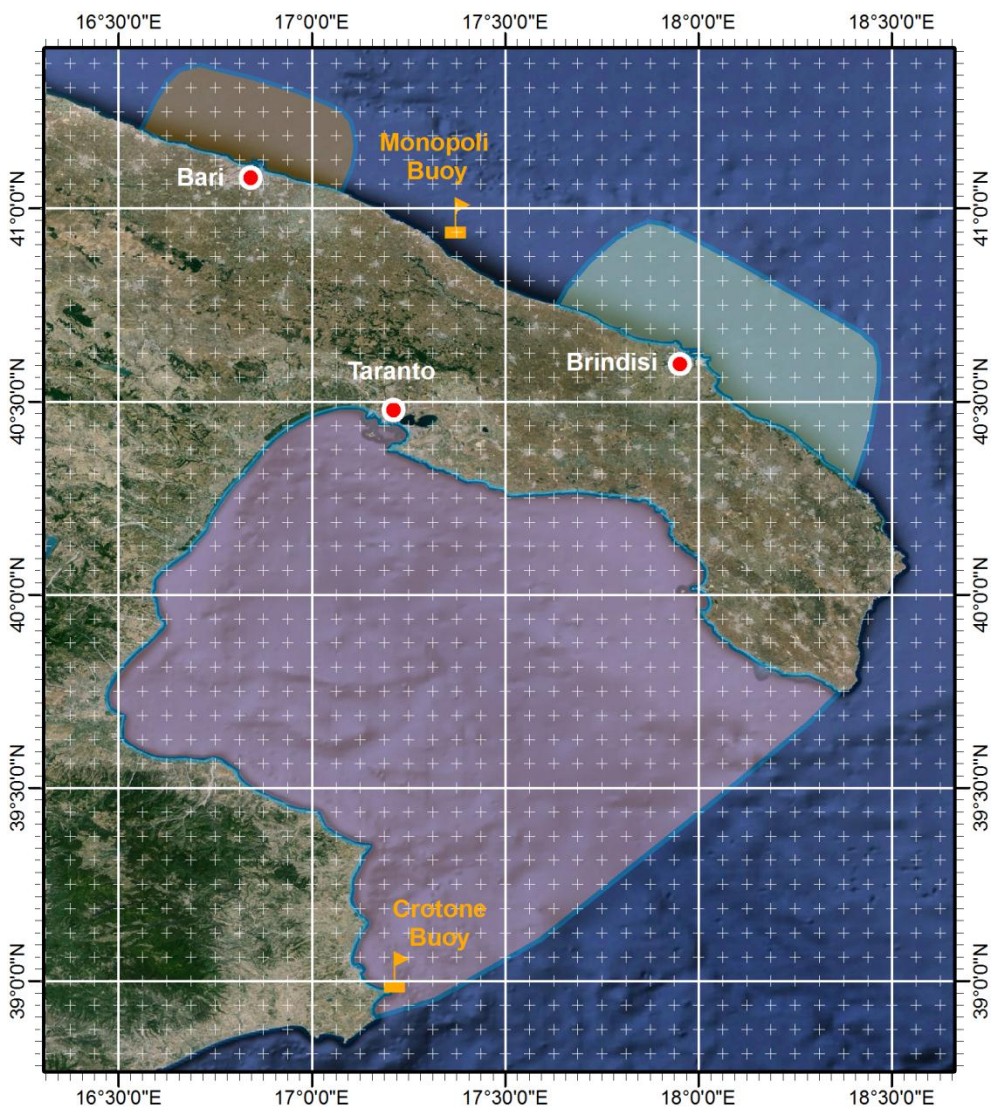





Figure 4. Frequencies of occurrence of the scenarios' $H_s$ - $T_p$ and $H_s$ - $Dir_m$ pairs for the Monopoli dataset ((a) and (b), respectively) and the Crotone dataset ((c) and (d), respectively).

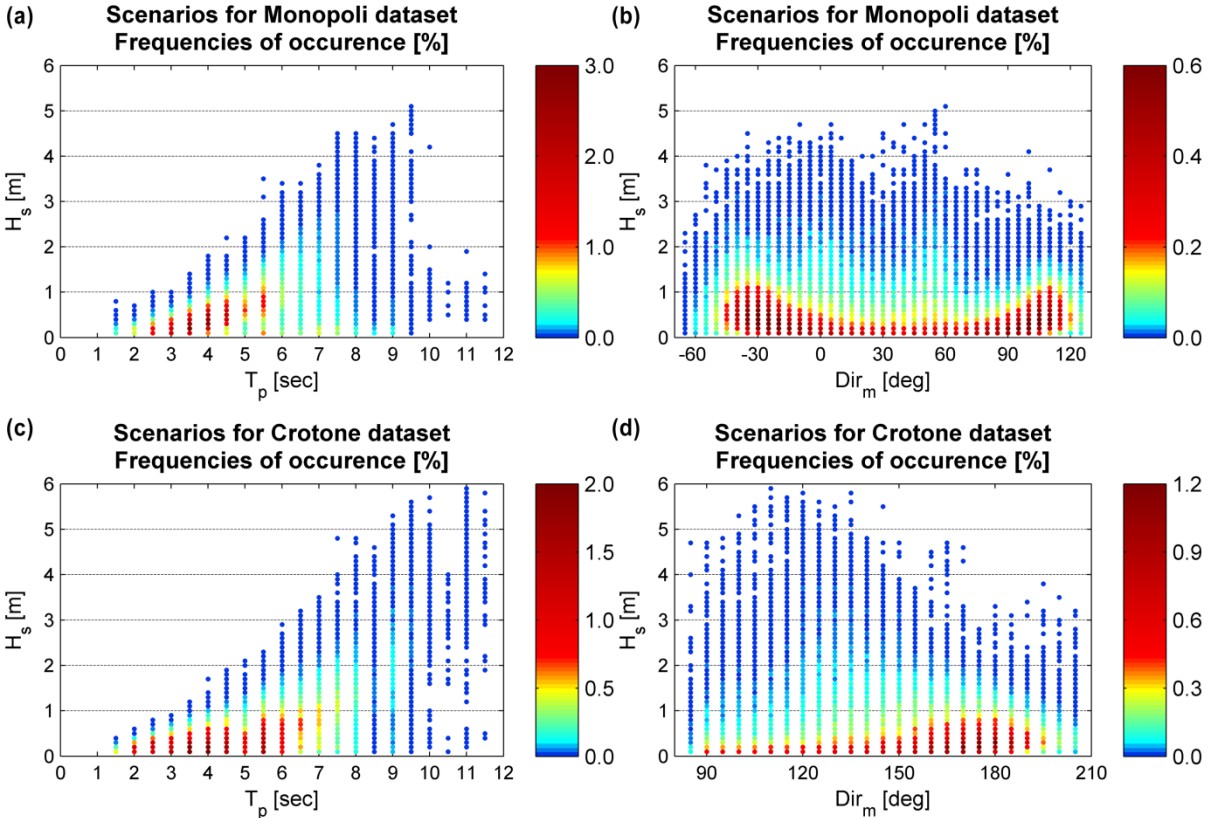





Figure 5. Comparison of TELEMAC and MIKE21 results ($H_s$ - $T_m$ - $Dir_m$) along trajectories: (a) *TRc1* and (b) *TRc2*, for the Brindisi – Torre Guaceto case study (forcing *WE1*; * = *Tc14*).





Figure 6. (a) Comparison of TELEMAC and MIKE21 results ($H_s$ - $T_m$ - $Dir_m$) along trajectory *TRc3* for the Brindisi – Torre Guaceto case study (forcing *WE1*; * = *Tc14*); (b) effect of different processes on $H_s$ for TELEMAC (top) and MIKE21 (bottom).



Figure 7. Comparison of TELEMAC and MIKE21 results ($H_s$ - $T_m$ - $Dir_m$) along trajectories: (a) *TRc1* and (b) *TRc2*, for the Brindisi – Torre Guaceto case study (forcing *WE2*; * = *Tc24*).



Figure 8. (a) Comparison of TELEMAC and MIKE21 results ($H_s$ - $T_m$ - $Dir_m$) along trajectory *TRc3* for the Brindisi – Torre Guaceto case study (forcing *WE2*; * = *Tc24*); (b) effect of different processes on $H_s$ for TELEMAC (top) and MIKE21 (bottom).





Figure 9. Comparison of TELEMAC and MIKE21 results ($H_s$) at points within (*PTc1*, *PTc2*, *PTc3*) and outside of the breaker zone (*PTc4*, *PTc5*, *PTc6*) for the Brindisi–Torre Guaceto case study for runs: (a) *Tc31*, (b) *Tc32* and (c) *Tc33/Tc34*\*.





Figure 10. Comparison of TELEMAC and MIKE21 results (*Curr. speed/direction*) at points within the breaker zone (*PTc1*, *PTc2*, *PTc3*) for the Brindisi–Torre Guaceto case study for runs: (a) *Tc31*, (b) *Tc32* and (c) *Tc33/Tc34*.



Figure 11. Comparison of TELEMAC and MIKE21 results ($H_s$) at points within (*PTc1*, *PTc2*, *PTc3*) and outside of the breaker zone (*PTc4*, *PTc5*, *PTc6*) for the Brindisi–Torre Guaceto case study for runs: (a) *Tc41*, (b) *Tc42* and (c) *Tc43/Tc44\**.






Figure 12. Comparison of TELEMAC and MIKE21 results (*Curr. speed/direction*) at points within the breaker zone (*PTc1*, *PTc2*, *PTc3*) for the Brindisi–Torre Guaceto case study for runs: (a) *Tc41*, (b) *Tc42* and (c) *Tc43/Tc44\**.





Figure 13. Comparison of TOMAWAC results for the Bari case study: (a), (d) along trajectory *TRh1* for forcings *WE1* and

*WE2*, respectively; (b)-(c) and (e)-(f) as wave fields at the harbour of Bari for forcings *WE1* and *WE2*, respectively.





Figure 14. Comparison of TOMAWAC results ($H_s$ - $T_m$ - $Dir_m$) for the Bari case study at points *PTh1*, *PTh2*, *PTh3* for forcings: (a) *TS1* and (b) *TS2*.