# Peer review of "High resolution wave and hydrodynamics modelling in coastal areas: operational applications for coastal planning, decision support and assessment"

_Natural Hazards and Earth System Sciences, 2016_

## Referee Comment (RC1) · C. Koutitas (Referee) · 11 Apr 2016

The paper is of excellent scientific importance and quality, relevant to the scientific targets of the periodical . It adresses a subject very interesting to the community of coastal oceanographers and coastal engineers, operationally important. The approach is clear and the applications illustrative of the validity and the comparative value of the tested wave models.

The paper has a quite excessive length and detailed descriptions that could be reduced a little to offer comfort to the reader.

[Figure]

From the technical point of view, the terms of the hydrodyamic model (aiming to the description of the waves generated currents) , should be clarified more (adding explicitely the radiations stresses terms, and their derivation).

It would be also interesting to have a qualitative or quantitative aspect of the authors on the importance of the wave generation and nonlinear interactions terms in the two applications, the first on narrow coastal strips and the second on the extend of a medium size coastal basin (bay).

No other technical corrections are recommended. The paper can be published as it is.

---

## Referee Comment (RC2) · Anonymous Referee #2 · 18 Apr 2016

The paper suggests an interesting approach for very high-resolution operational modelling for coastal systems with a broad range of applications, from shore protection to harbour design and management. The manuscript is properly organized and the work performed is very well described, to the advantage of the paper readability. Nevertheless, in order to fully exploit the paper potential, I suggest a minor revision aiming at consolidating a few points listed in the following. Also, as a non-native English speaker I could find some imprecisions suggesting that, although the paper is overall well written, a thorough check on the language is necessary.

[Figure]

One of the main points of the work lies in the comparison between MIKE21 and TELEMAC results, but there seems not to be a suitable observational background for evaluating the quality of the results and identifying which model actually provides better performances. I encourage either to discuss this point or to introduce some observational data in the analysis.

A slightly broader description of the differences between parameterisations in MIKE-SW and TOMAWAC, although they may be very small, would in my opinion significantly increase the insight on the physical implications of the differences between the model results.

If I properly understood, no atmospheric forcing is prescribed in the simulations performed. This choice should be justified, or at least corroborated by a sensitivity analysis exploring the impact on the results of including some (possibly idealized) atmospheric forcings (basically wind stress) in the simulation.

P3L8: Here and throughout the whole manuscript the authors refer to the domain sections on which the comparison is performed as "trajectories". This can sound somewhat misleading, as it might suggest that something is moving along those directions (suggesting in a way the idea of wave rays), therefore I would rather refer to "transects" or "lines".

P4L3: "breaking"

P5L13-16: I am not sure that showing the structure of the output file is necessary, and it is not clear to me what the files are actually bridging: are they not themselves a part of the dataset thet they are supposed to be linking to the coarser-resolution operational models?

P6L28: I suggest something like "posed by the inclusion of diffraction"

P6L29-31: again, the absence of a comparison against observed data can pose some serious limitations to the assessment of the model performances in this condition.

Please discuss.

P9L4: "MIKE"

---

## Author Comment (AC1) · 16 May 2016

Please find attached in the present, as separate *.pdf files in the compressed supplement, the document with the authors' response to the Referees' comments and the revision details, as well as the annotated revised manuscript.

We remain at your disposal should you need any further information.

Respectfully, A.G. Samaras, M.G. Gaeta, A. Moreno Miquel, and R. Archetti

Please also note the supplement to this comment:

http://www.nat-hazards-earth-syst-sci-discuss.net/nhess-2016-63/nhess-2016-63-AC1-supplement.zip